# Identification and Characterization of MicroRNAs in Gonads of *Helicoverpa armigera* (Lepidoptera: Noctuidae)

**DOI:** 10.3390/insects12080749

**Published:** 2021-08-19

**Authors:** Leyao Li, Shan Wang, Kaiyuan Huang, Yuting Zhang, Yalu Li, Min Zhang, Jinyong Huang, Zhongyuan Deng, Xinzhi Ni, Xianchun Li

**Affiliations:** 1School of Agricultural Sciences, Zhengzhou University, Zhengzhou 450001, China; lly0717@163.com (L.L.); sunnywang0507@163.com (S.W.); huang2420244465@163.com (K.H.); zhyuting2021@163.com (Y.Z.); liyalu2021@163.com (Y.L.); zhangmin753@gmail.com (M.Z.); hjy666@zzu.edu.cn (J.H.); 2School of Life Sciences, Zhengzhou University, Zhengzhou 450001, China; 3USDA-ARS, Crop Genetics and Breeding Research Unit, Tifton Campus, University of Georgia, Tifton, GA 31793, USA; xinzhi.ni@usda.gov; 4Department of Entomology and BIO5 Institute, University of Arizona, Tucson, AZ 85721, USA

**Keywords:** miRNAs, piRNAs, ovary, testis, reproductive destruction, development, sex

## Abstract

**Simple Summary:**

For most insects, the development of the testis and ovary directly determines their reproductive ability. The cotton bollworm, *Helicoverpa armigera* (Hübner), is a polyphagous crop pest of the Lepidoptera Noctuidae. Owing to its broad range of host plants and strong fertility, *H. armigera* causes huge economic losses to agricultural production. Acting as a type of post-transcriptional regulatory factor, miRNAs participate in the gonadal development and reproductive regulation of *H. arimgera*. Our study uses *H. armigera* as a research object to identify and characterize the miRNAs and study their potential functions in the testis and ovary of this destructive crop pest. A total of 7,592,150 and 8,815,237 clean reads were obtained by constructing small RNA libraries of the testis and ovary, respectively. Length distribution analysis showed that the main types of small RNAs in the testis and ovary were different. Among the 74 known miRNAs, 60 miRNAs existed in the ovary, and 72 existed in the testis. Gene Ontology (GO) and KEGG pathway analyses indicated that the 8 gonad-biased differentially expressed miRNAs (miR-989a, miR-263-5p, miR-34, miR-2763, miR-998, miR-2c, miR-2765, and miR-252a-5p) had many target transcripts involved in the reproduction process.

**Abstract:**

The high fecundity of the most destructive pest *Helicoverpa armigera* and its great resistance risk to insecticides and Bt crops make the reproductive-destruction-based control of this pest extremely appealing. To find suitable targets for disruption of its reproduction, we observed the testis and ovary development of *H. armigera* and conducted deep sequencing of the ovary and testis small RNAs of *H. armigera* and quantitative RT-PCR (RT-qPCR) validation to identify reproduction-related micro RNAs (miRNAs). A total of 7,592,150 and 8,815,237 clean reads were obtained from the testis and ovary tissue, respectively. After further analysis, we obtained 173 novel and 74 known miRNAs from the two libraries. Among the 74 known miRNAs, 60 miRNAs existed in the ovary and 72 existed in the testis. Further RT-qPCR validation of 5 miRNAs from the ovary and 6 miRNAs from the testis confirmed 8 of them were indeed ovary- (miR-989a, miR-263-5p, miR-34) or testis-biased (miR-2763, miR-998, miR-2c, miR-2765, miR-252a-5p). The 8 ovary- or testis-biased miRNAs had a total of 30,172 putative non-redundant target transcripts, as predicted by miRanda and RNAhybrid. Many of these target transcripts are assigned to reproduction-related GO terms (e.g., oocyte maturation, vitellogenesis, spermatogenesis) and are members of multiple reproduction-related KEGG pathways, such as the JAK-STAT signaling pathway, oocyte meiosis, the insulin signaling pathway, and insect hormone biosynthesis. These results suggest that the 8 gonad-biased miRNAs play important roles in reproduction and may be used as the targets for the development of reproductive-destruction-based control of *H. armigera* and, possibly, other lepidopteran pests.

## 1. Introduction

Non-coding RNAs (NcRNAs) refer to all functional RNA molecules that are transcribed from DNA but not translated into proteins. These include ribosomal RNA (rRNA), transfer RNA (tRNA), small nuclear RNA (snRNA), small nucleolar RNA (snoRNA), microRNA (miRNA), small interfering RNA (siRNA), piwi-interacting RNA (piRNA), and long non-coding RNA (lncRNA). Depending on their classes, ncRNAs may participate in one or multiple steps of the flow of genetic information from DNA to protein, including DNA replication, DNA methylation, chromatin remodeling, RNA transcription, RNA processing, transcriptional and post-transcriptional regulation, protein translation, and protein stability [1]. SnRNAs mainly act on the splicing of pre-mRNAs into mature mRNAs, tRNAs are involved in the recruitment of amino acids to generate peptide or protein, and rRNAs are the major components of ribosomes, the protein translation machinery [2]. SnoRNAs participate in the modification of snRNAs or rRNAs and are also related to the processing of rRNAs during ribosomal subunit maturation. LncRNAs can interact with DNA, RNA, and/or proteins and, thus, are involved in almost every step of the genetic information flow and regulation of many cellular processes [3]. Endogenous siRNAs function as the cellular defense mechanism against foreign nucleic acids [4]. PiRNAs play an important role in the silencing of transposons and the post-transcriptional regulation of mRNAs [5].

MicroRNAs are a class of non-coding single-stranded RNA molecules of 21–24 nucleotides (nt) in length that modulate the post-transcriptional regulation of target genes by interacting with the 3′ untranslated regions (UTR) of target mRNAs to induce mRNA degradation and translational repression [6,7,8]. However, recent research has shown that miRNAs can also interact with the 5′ UTR, coding sequences (CDS), and promoters of target genes [7,9]. In animals, miRNAs loaded with the argonaute protein can base-pair with target mRNA sequences in an incomplete manner to inhibit the translation of target mRNAs without reducing their stability [10]. MiRNAs can be secreted into extracellular fluids and transported to target cells via vesicles, such as exosomes, or by binding to proteins, including argonautes [7]. 

MicroRNAs have been shown to play key regulatory roles in the reproduction of insects [6,11]. Reproduction-related miRNAs often display a sex-biased expression pattern, suggesting that they play indispensable roles in reproduction processes, such as sex differentiation, gonadal development, oogenesis, and spermatogenesis [12,13]. In *Drosophila*, male-biased miRNAs are often enriched in testes, whereas female-biased miRNAs are expressed much higher in ovaries and early embryos [14]. The strongest ovary-biased *Drosophila* miRNA is miR-989 [15], which regulates border cell migration in the *Drosophila* ovary [16]; its homolog targets the key sex switch gene *doublesex* (*dsx*) in *Bactrocera dorsalis* [17]. MiR-318, the next strongest ovary-biased miRNA [15], is activated by 20-hydroxyecdysone (20E) to control eggshell chorion gene amplification and eggshell pattern formation with Tramtrack69 during *Drosophila* follicular epithelium differentiation and oogenesis [18]. Bantam, another ovary-enriched miRNA, is required for maintaining germline stem cells in the *Drosophila* ovary [19]. Analogous to miR-318 and Bantam in the *Drosophila* ovary, the testis-biased miRNAs miR-275 and miR-306 [15] downregulate Bag of Marbles (*Bam*), a key differentiation factor for the transition from proliferative spermatogonia to differentiating spermatocytes, to control the stem cell differentiation pathway and ensure proper spermatid terminal differentiation in *Drosophila* testes [20]. *Drosophila* miR-279 exhibits a complex expression pattern with testis-biased expression (testis/ovary = 4.5) in adult gonads, female-biased expression in L3 larvae salivary glands (M/F = 0.35), and unbiased expression in adult heads (M/F = 1.12) and adult bodies (M/F = 1.03) [15]. This testis-biased miRNA, together with Suppressor of Cytokine Signaling at 36E (Socs36E), attenuates the highly conserved Janus Kinase/Signal Transducer and Activator of Transcription (JAK/STAT) pathway, not only in the anterior follicle cells of *Drosophila* ovaries to optimize the number of border cells during oogenesis [21] but also in the stem cells of *Drosophila* testes to balance self-renewal and differentiation [22]. Apparently, such reproduction-related miRNAs, if identified from non-*Drosophila* agricultural pests, would be ideal targets for birth control or reproduction-disruption-based management of these pests [23,24,25,26]. Like miRNA-expressing [27,28] or RNAi (RNA interference) plants [29], crops can be genetically modified to produce reproduction-related miRNAs or their sponges, which, in turn, dramatically reduce the fecundity of the pests feeding on the crops or even sterilize them.

The cotton bollworm *Helicoverpa armigera* (Hǜbner) is one of the most destructive crop pests worldwide. High fecundity is among the three major factors (polyphagy, long-distance mobility, strong fertility) that have led to the success of *H. armigera* as a pest of agroecosystems [30]. However, management of this pest heavily relies on the toxicity of Bt toxins on Bt-transgenic crops [31,32] or insecticides on conventional crops [33]. Reproduction-disruption-based control tactics have yet to be developed for this devastating pest. Identification of the ovary- or testis-biased miRNAs of *H. armigera* would facilitate the development of such control tactics. Although the identification of *H. armigera* miRNAs has been previously reported, the distribution of miRNAs in the testis and ovary is still unknown [34]. To that end, we conducted high-throughput sequencing of small RNA samples extracted from male testes or female ovaries of *H. armigera*. The resultant known or novel *H. armigera* miRNAs were annotated with the publicly available miRNA sequences from related insect species, such as *Bombyx mori*, in the miRBase database. Among the known miRNAs, five from the ovary and six from the testis were further validated by RT-qPCR. We also bioinformatically predicted the putative target transcripts of the validated three ovary- and five testis-biased mRNAs. Taken together, we identified three ovary- and five testis-biased miRNAs, all of which have target genes involved in the sexual reproduction of *H. armigera*.

## 2. Materials and Methods

### 2.1. Insect Rearing

The *H. armigera* strain used in this study was obtained from KEYUN Biological Pesticide Co. Ltd in Jiyuan county, Henan province, China, in September 2018. Larvae were reared in 1 oz plastic cups (about 10 larvae/cup initially and 1 larva per cup from the 3rd instar) containing wheat-germ-based artificial diets and maintained at 28 °C, 60% RH, and a light–dark 16:8 photoperiod. Adults were reared in cages with 10% sugar solution.

### 2.2. Gonad Observation and Collection of Tissue Samples

To observe gonad development, we tried to dissect male testes and female ovaries from 4th instar larvae (L4), L5, L6, pre-pupa, pupa, newly emerged (0-day-old) adults, 1-day-old adults (females only), and 2-day-old adults (females). All testes and ovaries that were visible to the naked eye were individually observed and photographed in phosphate-buffered saline (PBS) with a hyper-depth 3D microscope (Smart-254 zoom5 1810070S, Zeiss).

Testis tissues from *H.armigera* were fixed in Bouin’s buffer for 1 h and stored in 70% (*v*/*v*) ethanol. The tissue was dehydrated and embedded in paraffin during sectioning. The 6 μm thick sections were deparaffinized with xylene and rehydrated with an ethanol concentration gradient from 100% (*v*/*v*) to 50% (*v*/*v*). The glass slides were soaked in distilled water for 5 min and dyed with hematoxylin at room temperature for 30 s. Then, the slides were soaked in distilled water again for 5 min, changing the water several times during this step. Subsequently, the slides were washed for several seconds in ammonium hydroxide to make the stains blue before being stained with eosin for 30 s and dehydrated with an ethanol concentration gradient from 50% (*v*/*v*) to 100% (*v*/*v*). Photos were captured using a microscope (CX53, Olympus, Japan).

We prepared one male testis sample and one female ovary sample for Illumina sequencing of the testis and ovary small RNAs, respectively. The male testis sample contained 6 testes, which were dissected out from 6 sexually mature (i.e., newly emerged) male adults, flash-frozen in liquid nitrogen, and stored at −80 °C before RNA extraction. The female ovary sample included 6 pairs of ovaries, which were dissected from 6 sexually mature (i.e., 2-day-old) female adults, flash-frozen in liquid nitrogen, and stored in the same ultracold freezer as the male testis sample. 

For quantitative RT-PCR (RT-qPCR) validation of the 6 candidate testis-biased miRNAs and the 5 candidate ovary-biased miRNAs revealed by Illumina sequencing, 3 biological replicates of 3 testes (or pairs of ovaries) each were dissected out from newly-emerged male moths and 2-day-old female moths, respectively, and prepared in the same way as the two gonad samples for Illumina sequencing. For RT-qPCR analyses of the spatial expression profiles of the 8 validated sex-biased miRNAs, 3 replicates of heads, thoraxes, abdomens, legs, wings, fat bodies, testes, and ovaries were dissected from the same set of 2-day-old male and female adults, respectively. Each replicate of the 8 tissues or body parts contained the corresponding tissue or body parts from a male or female moth. These tissue/body part samples were flash-frozen in liquid nitrogen and store at −80 °C before RNA extraction. 

### 2.3. RNA Extraction and Library Construction

Each of the aforementioned tissue or body part samples was ground in liquid nitrogen, and total RNA was extracted using the TRIzol reagent (TaKaRa), according to the manufacturer’s protocol. The quality and quantity of total RNA were examined by a NanoDrop ND-2000 spectrophotometer (Thermo Fisher Scientific).

About 10 μg of the obtained testis and ovary total RNA samples were sent to Novogene (Beijing, China) for construction and sequencing of the testis and ovary small RNA libraries. Briefly, 10 μg of the testis and ovary total RNA samples were fractioned on 15% denaturing polyacrylamide gel electrophoresis (PAGE) gel. Small RNAs of 15–50 bp were eluted from the corresponding gel slices. The resultant clean small RNAs were then ligated to one 3’ adaptor and one 5’ adaptor using T4 RNA ligase. The adaptor-ligated small RNAs were reverse-transcribed into cDNAs, which were further PCR-amplified to yield the testis and ovary libraries. The two libraries were purified by denaturing PAGE and then sequenced (50 bp single-end) on an Illumina HiSeq 2000 machine platform.

### 2.4. Small RNA Sequence Analysis

The original data, obtained using an Illumina sequencing analyzer, were transformed into raw read sequences using base calling and stored as FastQ format files. The read sequence data quality was checked using FastQC (version 0.11.3). After removing the adaptor sequences of the reads with Cutadapter (version 1.18), including trimming the 3′ and 5′ adaptors, low-quality reads with Trimmomatic (version 0.36), and reads shorter than 15 nt or longer than 50 nt, the clean and filtered reads were obtained for further analysis. All the clean reads of the two libraries have been submitted to the SRA (Sequence Read Archive) database of NCBI (accession No. PRJNA613606).

These clean reads were annotated into different RNA classes, including rRNA, tRNA, snRNA, snoRNA, and miRNA, by Bowtie against the Rfam database (http://rfam.xfam.org/, version 14.3, accessed date 20 December 2020). The clean reads that are 26–31 nt in length and do not belong to the above 5 classes of small RNAs were further Blast-searched against the piRNA database (piRBase) to identify known piRNAs. The piRNA databases contain known piRNAs from human, mouse, rat, *D. melanogaster*, *C. elegans*, zebrafish, chicken, *X. tropicalis*, silkworm, starlet sea anemone, cow, crab-eating macaque, rhesus, marmoset, sea hare, tree shrew, pig, *D. erecta*, *D. yakuba*, *D. virilis*, and rabbit. The remaining reads that did not hit a piRNA in the piRNA database but had a uridine base (U) at the first position were considered novel piRNA.

### 2.5. Identification and Expression Analysis of miRNAs

Known and novel miRNAs were mainly discovered by miRDeep2 software (version 2.0.0.8) with default parameters (full length, no mismatches, randfold *p*-value ≤ 0.05). Due to the incomplete information on the transcriptome and genome of *H. armigera*, all the clean reads were aligned to the mature miRNAs and hairpin sequences of all species, particularly the model lepidopteran *Bombyx mori*, deposited in the miRBase (http://www.mirbase.org/, version 22.1, accessed date 1 August 2020), to identify known miRNAs using miRDeep2 (full length, no mismatches, miRDeep2 score > −10, randfold *p*-value ≤ 0.05). The reads with no significant hits were used to predict novel miRNAs using the RNAfold algorithm of miRDeep2 (version 2.0.0.8). The predicted precursors were considered novel miRNAs.

The read numbers mapped to each identified known and novel miRNA were counted for both the ovary and testis libraries and normalized to per million mapped reads using HTSeq v0.5.3 (EMBL, Heidelberg, Germany). MiRNAs with |log_2_Fold Change (FC)| ≥ 1 in the normalized read counts between the two libraries were considered gonad-biased candidate miRNAs. Heatmaps based on the log_2_FC values of miRNAs were drawn to show their putative relative expression levels in the ovary and testis with TBtools (v0.66839). 

### 2.6. RT-qPCR Validation of 11 Gonad-Biased Candidate miRNAs 

PrimeSriptTM RT Reagent Kit with gDNA Eraser (TaKaRa) and miRNA or reference small RNA specific stem-loop RT primers (Appendix A) were used to reverse-transcribe the corresponding candidate miRNA or reference small RNA in 1 μg of each total RNA sample into the corresponding cDNA. The program was as follows: incubation at 42 °C for 5 min to remove genome DNA, and 1 cycle of 42 °C for 15 min and 85 °C for 5 s to complete the reverse transcription reaction. Roche LightCycler 480 was used to perform real-time quantitative PCR (qPCR) with FastStart Universal SYBR Green Master (Roche). Each 20 μL qPCR reaction contained 10 μL SYBR Green I Master, 2 μL cDNA (10× dilution with ddH_2_O), 1 μL forward primer, 1 μL reverse primer, and 6 μL ddH_2_O. The qPCR circling program was 5 min at 95 °C for pre-denaturation, followed by 40 cycles at 95 °C for 10 s, 56 °C for 10 s, and 72 °C for 15 s, and a final extension step at 72 °C for 10 min. Three biological replicates and three technical replications were carried out. In order to obtain more credible results, multiple reference genes should be used to normalize expression [35]. U6 snRNA and let-7 were used as two internal reference small RNAs to normalize the relative abundance of each candidate miRNA [34,35,36]. 

The relative expression levels of the tested miRNAs were calculated with the 2^−ΔΔCT^ method [35]. Two sample *t*-tests were conducted to determine if the expression of a tested miRNA differed significantly between testis (or male for other tissues/ body parts) and ovary (or female for other tissues/body parts). The results are displayed in histograms using GraphPad Prism 5.0 software (GraphPad Software Inc., San Diego, CA, USA). 

### 2.7. Target Prediction and Enrichment Analysis of 8 Validated Gonad-Biased miRNAs

A comprehensive transcriptome dataset of *H. armigera*, assembled by Zhang [37], was used to predict the putative target transcripts of 8 validated gonad-biased miRNAs. This transcriptome dataset was merged with 21 (17 publicly available) RNA-seq read data of various stages, tissues, and body parts of *H. armigera* and 30,532 publicly available EST sequences (up to October 2015) and contained a total of 129,027 transcripts [37]. Annotations of the 129,027 transcripts were performed by comparing them with the NR, String, Swissprot, and KEGG databases using BlastX (BLAST version 2.2.25, E value < 10^−^^5^) and presented in Dataset S1. Two miRNA target prediction software, miRanda (version 3.3a) and RNAhybrid (version 2.1.2), were used to hunt for putative target transcripts of 8 validated gonad-biased miRNAs. According to the principle of 5′ seed primacy in miRNA-target binding, any of the 129,027 transcripts predicted to have an uninterrupted 7 nt strong complementary base-pairing site to the 5′ seed region (nucleotide position 2–8) of a given miRNA were considered the putative target transcripts of the corresponding miRNA. The parameters used to screen for such complimentary sites were score > 140 and energy < −83.68 KJ/mol for miRanda prediction, and energy < −83.68 KJ/mol and *p*-value ≤ 0.05 for RNAhybrid prediction. Only the transcripts predicted by both software were considered the putative targets of a miRNA. Gene ontology (GO) term and Kyoto Encyclopedia of Genes and Genomes (KEGG) pathway enrichment analyses were carried out to infer if the predicted non-redundant putative target transcripts of the 8 gonad-biased miRNAs were enriched in the reproduction-related molecular functions, cell components, biological processes, and KEGG pathways using OmicShare Tools. The GO term and KEGG pathways with *p*-values ≤ 0.05 were considered to be significantly enriched among the predicted non-redundant putative target transcripts.

### 2.8. Data Analysis

We conducted two sample *t*-tests to compare the expressional differences of the 11 candidate miRNAs between ovary and testis tissues. Significant differences at *p* < 0.05, *p* < 0.01, and *p* < 0.001 are depicted with 1, 2, and 3 asterisks, respectively. All statistical analyses were performed in SPSS version 19.0 (SPSS Inc., Chicago, IL, USA).

## 3. Results

### 3.1. Gonadal Development

The high fertility of *H. armigera* is closely related to the normal development of the gonads. We examined the development of the male testis and female ovary in L4, L5, L6, pre-pupa, pupa, 0-day-old adults, 1-day-old adults (females only), and 2-day-old adults (females only). Male larvae had two orange petal-shaped testes, which gradually fused to form a spheroid in the pre-pupal stage (Figure 1A–D). From L4 to the early pre-pupal stages, the two testes remained separate (Figure 1A–D). By the end of the pre-pupal stage, the two testes began to approach each other and gradually merged into a spherical testis after pupation (Figure 1E–G). In newly emerged (0-day-old) male adults, the spherical testis is fully matured, and, at this point, the males are ready to mate with the females (Figure 1H). This was evidenced by completion of both the external (Figure 1A–H) and internal fusion (Figure 1I–K) of the two bilaterally symmetrical testes and of their eight follicles into one testis of a single large follicle, as well as the presence of mature sperm bundles (Figure 1K). During the whole process, the inside of the testes underwent great changes, including the disappearance of the septa between the two testes and between the follicles within each testis, the mitosis of spermatogonia into spermatocytes, the meiosis of spermatocytes into spermatid, and the extension and full differentiation of spermatid into mature sperm bundles.

By contrast, the ovaries of *H. armigera* developed much more slowly and, thus, were not visible to the naked eye from L4 to pre-pupae (data not shown). It became visible starting from the pupal stage (Figure 1A’) and grew rapidly after emerging as adults (Figure 1B’–D’). In 2-day-old female moths, the symmetrical pair of ovaries were large enough to fill the entire abdominal cavity, and both the lateral and common oviducts were full of mature eggs (Figure 1D’). Because male and female *H. armigera* sexually matured at emergence and 2 days post-emergence, respectively, we used testes from 0-day-old male adults and ovaries from 2-day-old female moths to identify testis- and ovary-biased miRNAs.

### 3.2. Analysis of Small RNA Sequencing Data

In order to identify miRNAs in the ovary and testis, we sequenced two small RNA libraries from the ovary and testis of *H. armigera* using the Illumina Hiseq platform. After removing adaptors, low-quality reads, and reads longer than 35 nt or shorter than 18 nt, the ovary and testis libraries contained 7,592,150 and 8,815,237 clean reads, respectively (Table 1). The length distributions of small RNAs showed one 20–24 nt peak in the testis and one 26–27 nt peak in the ovary (Figure 2). This suggested that the main types of small RNAs in the testes and ovaries of *H. armigera* were different. The 20–23 nt small RNAs in the testes represent the miRNAs, which play a crucial role during post-transcriptional regulation. However, the 26–27 nt small RNAs were largely piRNAs, which usually function in germline development.

The clean reads were divided into different types of small RNAs, including rRNA, tRNA, snRNA, snoRNA, unannotated RNAs, other RNAs, and piRNAs by comparing them with those in the Rfam database or the piRNA database (Table 1 and Figure 3). In both libraries, the ratios of rRNA (11.1% in ovary and 15.65% in testis), miRNA (11.85% in ovary and 15.45% in testis), and piRNA (34.40% in ovary and 11.49% in testis) were more than 10%. Overall, miRNAs have a higher proportion in testes, and piRNAs have a higher proportion in ovaries (Table 1 and Figure 3). 

### 3.3. Identification and Expression of miRNAs in the Testis and Ovary from H. armigera

Comparisons were made with known miRNAs in the miRBase database, and miRDeep2 was used to predict new miRNAs. If a miRNA perfectly mapped (full length, no mismatches) to one previously reported miRNA hairpin in the database, it was considered a known miRNA and named accordingly. Otherwise, it was assigned as a novel miRNA. In total, we identified 74 known miRNAs, of which 14 were present only in the testis library, 2 were present only in the ovary library, and 58 were shared by both libraries (Figure 4A). We also found 173 novel miRNAs: 72 in the ovary library, 64 in the testis library, and 37 in both the ovary and testis libraries (Figure 4B). All the known and novel miRNAs had a length of 21–24 nt (Appendix A), which is the typical size of Dicer processing products [38]. Additionally, 58.1% of these miRNAs had uracil (U) at their 5′ end (Appendix A), consistent with the nucleotide bias of miRNA at their first position [39].

To preliminarily screen for putatively sex-biased miRNAs between the mature ovary and testis of *H. armigera*, we compared the expression levels (i.e., fold change (FC) in the normalized read counts between the ovary and testis libraries) of all the identified known miRNAs. Among the 74 known miRNAs, 60 could be considered sex-biased candidate miRNAs because they had a |log2FC| of ≥1 between the male testis and female ovary (Figure 5). Interestingly, only 10 of the 60 candidate miRNAs were seemingly ovary-biased, whereas 50 of them were presumably testis-biased. 

### 3.4. Validation of 11 Sex-Biased Candidate miRNAs Using RT-qPCR

In order to identify gonad-biased miRNAs and verify the reliability of the sequencing results, we analyzed the expression levels of five ovary-biased (miR-989a, miR-34, miR-263a-5p, miR-1a-5p, and miR-263b-5p) and six testis-biased (miR-2c, miR-998, miR-252a-5p, miR-31-5p, miR-2765, and miR-2763) candidate miRNAs (Figure 5) by RT-qPCR, respectively. Three (miR-989a, miR-34, and miR-263b-5p) of the five ovary-biased candidate miRNAs revealed by Illumina sequencing were found significantly upregulated in the ovary, but two (miR-263a-5p and miR-1a-5p) of them were not biased (Figure 6). Among the six testis-biased candidate miRNAs, five (miR-2c, miR-252a-5p, miR-998, miR-2763, and miR-2765) were significantly upregulated in testis and only miR-31-5p was equally expressed between the testis and ovary (Figure 6). Correlation analysis between the male/female (M/F) ratios of the normalized Illumina sequencing read numbers, and the RT-qPCR expression levels of the 11 gonad-biased candidate miRNAs yielded a high correlation coefficient (R2 = 0.733) (Figure 7), indicating that our sequencing data are reliable. 

To determine if the eight validated ovary- or testis-biased miRNAs were also sex-biased in other tissues or body parts, we further analyzed their expressional levels in the head, thorax, abdomen, leg, wing, and fat body of 2-day-old male and female adults, respectively (Appendix A). Among the three ovary-biased miRNAs, miR-263b-5p was female-biased in all the six tested non-gonad tissues except for the wing, whereas miR-34 displayed no sexual difference in all the six non-gonad tissues (Appendix A). The remaining ovary-biased miR-989a was female-biased in the thorax and wing but not in the other four tissues. Among the five testis-biased miRNAs, miR-252a-5p and miR-2765 were not sex-biased in any of the six non-gonad tissues. Somewhat surprisingly, the testis-biased miR-2763 was female-biased in the wing, and the testis-biased miR-998 was female-biased in both the abdomen and wing. The testis-biased miR-2c was male-biased in the head but female-biased in the wing (Appendix A).

### 3.5. Target Prediction and Function Analysis

Using miRanda and RNAhybrid software, we predicted the target transcripts of the eight validated ovary- or testis-biased miRNAs mentioned above. For each of the eight miRNAs, only the transcripts predicted by both software were considered the putative targets of that miRNA. The number of putative target transcripts per miRNA varied dramatically, ranging from 324 for miR-2763 to 12,041 for miR-34 (Table 2). About 26.5% (10,882) of the putative target transcripts were shared by two or more of the eight miRNAs. When each of the shared putative target transcripts was counted just once, the eight testis- or ovary-biased miRNA had a total of 30,172 non-redundant putative target genes (Table 2), which were annotated by BlastX by searching for them against the NR, String, Swissprot, and KEGG databases. Gene Ontology (GO) functional category analysis assigned the 30,172 putative target transcripts to 67 level-2 GO terms, including reproduction and the reproduction process (Figure 8A). Notably, 7564 and 7404 of the 30,172 putative target transcripts participate in reproduction and the reproductive process, respectively (Figure 8A). KEGG pathway analysis showed that the 30,172 putative target transcripts belong to 42 level-2 (i.e., B class) categories (Figure 8B). At least 5 of the 42 B-class categories, including cell growth and death (843 transcripts), signaling transduction (2595 transcripts), lipid metabolism (748 transcripts), metabolism of terpenoids and polyketides (85 transcripts), and the endocrine system (1528 transcripts), were involved in reproduction (Figure 8B). By further examination of the five reproduction-related B class categories, we found that many of these putative target transcripts were involved in multiple reproduction-related pathways, including oocyte meiosis (*Ko04114*, 411 transcripts), the mTOR signaling pathway (*Ko04150*, 653 transcripts), the JAK-STAT signaling pathway (*Ko04630*, 166 transcripts), steroid hormone biosynthesis (*Ko00140*, 285 transcripts), the biosynthesis of insect hormones (*Ko00981*, 62 transcripts), and ovarian steroidogenesis (*Ko04913*, 100 transcripts) (Table 3).

## 4. Discussion

The high fertility of *H. armigera* and its ability to rapidly develop resistance to insecticides [36,40,41] and Bt crops [42,43] make the reproductive-destruction-based control of this pest very attractive. To find suitable reproduction-related targets for making this pest sterile or significantly reducing its fecundity, we first observed the gonad development of *H. armigera*. As in the lepidopteran insects *Helicoverpa zea* [44], *Spodoptera litura* [45], *Agrotis yprilon* [46], and *Ostrinia nubilalis* [47], male larvae of *H. armigera* possess two well-separated, petal-shaped testes, which gradually fuse to form a spheroid in the pre-pupal stage (Figure 1) [48]. However, no testicular fusion occurs during the testis development of *Bombyx mori* [49], the domesticated model lepidopteran insect. This observation also led to the determination of the sexually matured stages for finding reproduction-related targets, i.e., newly emerged male adults for testes and 2-day-old female adults for ovaries (Figure 1). Consistent with our finding, 2-day-old *H. armigera* female moths are known to carry ovaries of the egg maturation phase, with plenty of eggs in alignment [50].

We then conducted small RNA sequencing and RT-qPCR validation to identify reproduction-related miRNAs, with total RNA samples extracted from the testes of newly emerged male adults and the ovaries of 2-day-old female moths. This is because miRNAs are known to play important roles in the various steps of insect reproduction [6,11,51], such as spermatogenesis [52], oogenesis [11,51], and sex determination [53]. A total of 244 miRNAs were found in the gonads of the sexually matured moths of *H. armigera* (Figure 4), whereas only 107 miRNAs were previously identified from the whole bodies of various stages of this pest [34]. Furthermore, three ovary- (miR-263b-5p, miR-989a, miR-34) and five testis-biased (miR-2763, miR-252a-5p, miR-998, miR-2765, miR-2c) miRNAs (Figure 6) were identified from a total of 7,592,150 and 8,815,237 clean reads from the testis and ovary libraries of *H. armigera* (Table 1). Consistent with our finding, miR-989 (i.e., miR-989a) has been found to be an ovary-biased miRNA in all the examined insects, including the dipterans *D. melanogaster* [14,15], *Aedes aegypti* [54], *Anopheles anthropophagus* [55], *Anopheles gambiae* [56], and *B. dorsalis* [17] and the lepidopteran *Pararge aegeria* [57]. Like in *H. armigera*, miR-34 is ovary-biased in Reeves’ pond turtle *Mauremys reevesii* [58]. Interestingly, miR-263b, another ovary-biased miRNA in *H. armigera*, is ovary-biased in the cultivated oyster *Crassostrea hongkongensis* [59] and the mud crab *Scylla paramamosain* [12] but is testis-biased in *D. melanogaster* [15], even though the latter is evolutionarily closer to *H. armigera*. Among the five testis-biased miRNAs in *H. armigera*, miR-252 and miR-2c are also more abundant in *Drosophila* testis than in *Drosophila* ovaries, but miR-998 is enriched in the ovary of *D. melanogaster* [15]. Along the same line, miR-2763 is ovary-biased rather than testis-biased in *P. aegeria* [57]. The consistent gonad bias of miR-989, miR-252, and miR-2c across species is probably a reflection of their more important roles in female reproduction than male reproduction. By the same token, the species-specific gonad bias of miR-263b, miR-998, and miR-2763 may indicate that their relative contributions to male vs. female reproduction are species-dependent. Alternatively, the species-specific gonad bias of the three miRNAs is simply due to different gonad collection times, such as newly emerged males and 2-day-old females for *H. armigera* (this study) and 1-day-old males and females for *D. melanogaster* [15]. Future RT-qPCR analysis of the temporal relative abundances of the three miRNAs in the ovary and testis of the two species will resolve the two possibilities.

The aforementioned eight gonad-biased miRNAs are surely just a portion of the gonad-biased miRNAs because our analysis of sequencing data found 10 ovary- and 50 testis-biased candidate miRNAs (Figure 5). Further RT-qPCR validation of the remaining gonad-biased candidate miRNAs is needed to identify other reproduction-related miRNAs. Our findings of more testis-biased miRNAs than ovary-biased miRNAs (Figure 5 and Figure 6), as well as of more piRNAs in the ovary but more miRNAs piRNAs in the testis (Figure 2), may suggest that miRNAs are more involved in testis development and spermatogenesis, whereas piRNAs play bigger roles in ovary development and oogenesis. This notion is consistent with the function of piRNAs and the sex chromosome composition of male and female *H. armigera* and other lepidopterans. As the largest group of small non-coding RNAs in animal gonads, piRNA protects the germline genome and gametogenesis by inhibiting transposons [60]. In males, there are two *Z* chromosomes (*ZZ*), both of which, like autosomes, contain a large number of protein-coding genes and some transposons. By contrast, female lepidopterans are heterogametic (*ZW*), containing one *Z* chromosome and one *W* chromosome. Unlike the *Z* chromosome, the *W* chromosome is replete with transposons but poor in protein-coding genes [61,62]. This suggests that the lepidopteran ovary contains more transposons and, thus, requires more piRNAs, some of which may be derived from the *W* chromosome [63] to defend its germline genome against the extra numbers of transposons.

The eight ovary- or testis-biased miRNAs had a total of 30,172 non-redundant putative target transcripts (Table 2). GO functional category analysis revealed that over 7500 of these putative target transcripts participate in reproduction and the reproductive process, respectively (Figure 8A). The involved specific reproduction-related GO terms include sex determination (*GO:0007530*), primary ovarian follicle growth (*GO:0001545*), Sertoli cell proliferation (*GO:0060011*), meiotic cell cycle process involved in oocyte maturation (*GO:1903537*), vitellogenesis (*GO:0007296*), spermatogenesis (*GO:0007283*), and genitalia development (*GO:0048806*). From the above results, it can be concluded that these miRNAs regulate reproductive development and sex determination from many aspects. In line with our GO functionary analysis, our KEGG analysis also indicated that many of these putative target transcripts are members of the KEGG pathways related to reproduction, gonadal development, and hormone regulation, which include the JAK-STAT signaling pathway (*Ko04630*), oocyte meiosis (*Ko04114*), insect hormone biosynthesis (*Ko00981*; including JH and 20E), steroid hormone biosynthesis (*Ko00140*), and the insulin signaling pathway (*Ko04910*) (Figure 8B and Table 3). The JAK-STAT signaling pathway is a signal transduction pathway stimulated by cytokines which are involved in many important biological processes such as cell proliferation, differentiation, apoptosis, and immune regulation. In *Drosophila*, the JAK-STAT signal regulates the self-renewal of stem cells in spermatogenesis [64]. Various hormones, such as JH and 20E in insects, synergistically regulate insect metamorphosis and reproduction development. JH is synthesized in corpora allata to regulate the yolk development of most insects and promotes the development of the adult ovary. In lepidopterans, JH promotes vitellogenin uptake by ovaries. 20E is the main hormone that regulates female reproduction in some Hymenoptera, Lepidoptera, and all Diptera [11]. According to a previous study on insulin signaling, evidence showed that insulin signaling facilitates previtellogenic development and enhances JH-mediated vitellogenesis in *Maruca vitrata* [65]. In general, the results of both GO and KEGG analyses confirm the potential roles of the eight gonad-biased miRNAs in sex determination, gonadal development, and gametogenesis. Therefore, field application or in-planta production of the agomirs and/or antagomirs of the eight gonad-biased miRNAs is expected to make *H. armigera* sterile and/or lay fewer eggs.

## 5. Conclusions

We constructed two small RNA libraries of *H. armigera* testis and ovary to reveal testis- and/or ovary-expressed miRNAs. Among the known miRNAs, 50 of them were putatively upregulated in the testis, and 10 were seemingly upregulated in the ovary. Eight of them were validated by RT-qPCR to be testis- or ovary-biased miRNAs. The eight gonad-biased miRNAs may play important roles in various biological processes of *H. armigera* growth and development, including gonadal development and reproductive regulation.

## Figures and Tables

**Figure 1 insects-12-00749-f001:**
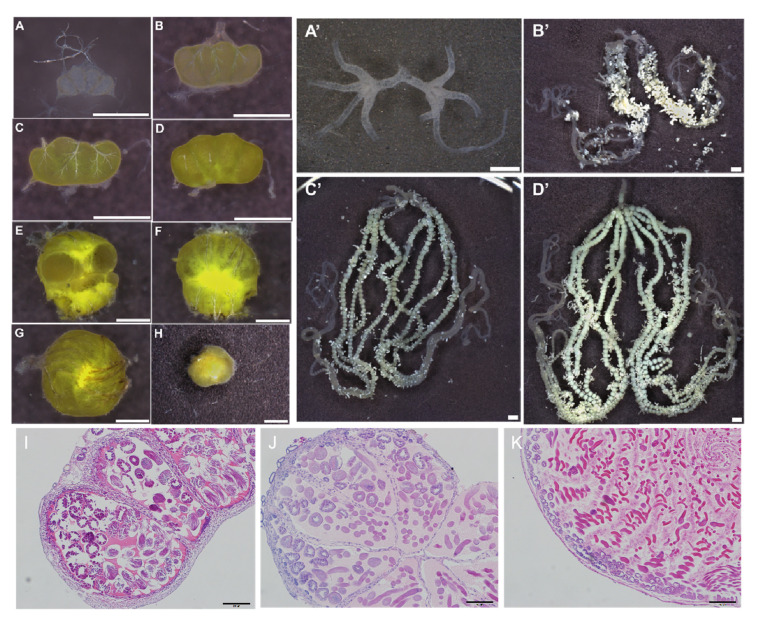
Development of *H. armigera* testis and ovary. (**A**–**H**) The testes of L4, L5, L6, pre-pupa, pupa, and newly emerged (0-day-old) male adults, respectively. (**A’**–**D’**) The ovaries of pupa, newly emerged (0-day-old) female adults, 1-day-old female adults, and 2-day-old female adults, respectively. Bar = 1 mm. (**I**–**K**) The testis tissue of larvae, late pupal stage, and newly emerged adults, respectively, by H&E staining. Bar = 100 μm. (**A**–**H**,**A’**–**D’**) Photos taken by a hyperdepth 3D microscope (Smartzoom5 1810070S, Zeiss). Photos of ovaries were stitched by microscope.

**Figure 2 insects-12-00749-f002:**
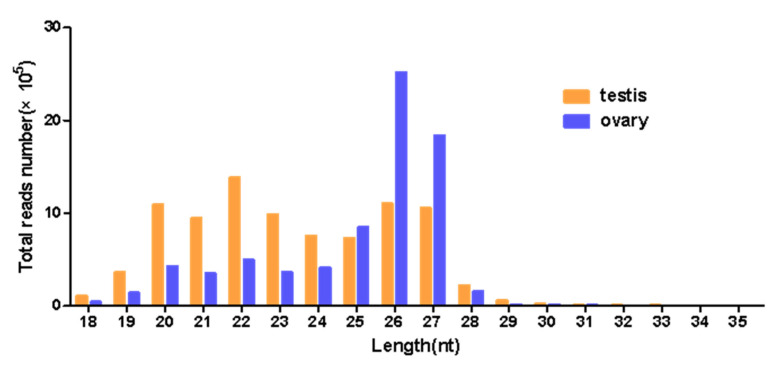
The length distribution of small RNAs in the testis of a 0-day-old male adult and the ovary of a 2-day-old female adult of *H. armigera*.

**Figure 3 insects-12-00749-f003:**
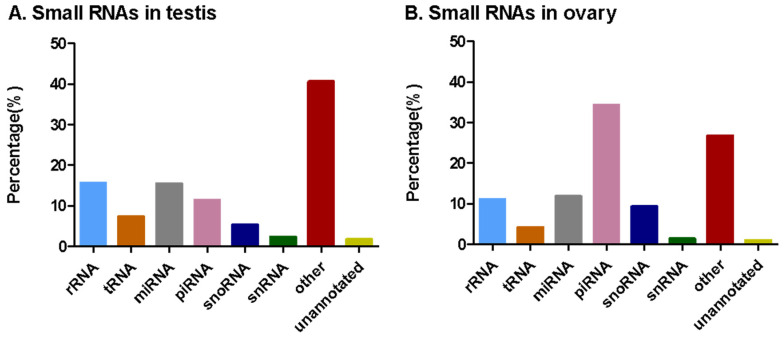
Composition of different types of small RNAs in the testes (**A**) and ovaries (**B**) of 0-day-old males and 2-day-old female adults of *H. armigera*. Small RNAs were annotated by the Blast search against the Rfam and piRBase databases. The range length for small RNAs is 15–50 bp.

**Figure 4 insects-12-00749-f004:**
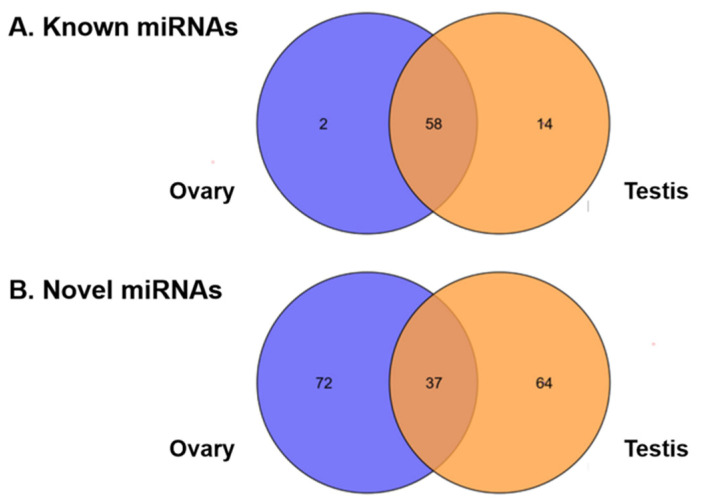
The numbers of the shared and testis- or ovary-specific known (**A**) and novel (**B**) miRNAs expressed in the gonads of a 0-day-old male adult and a 2-day-old female adult of *H. armigera*.

**Figure 5 insects-12-00749-f005:**
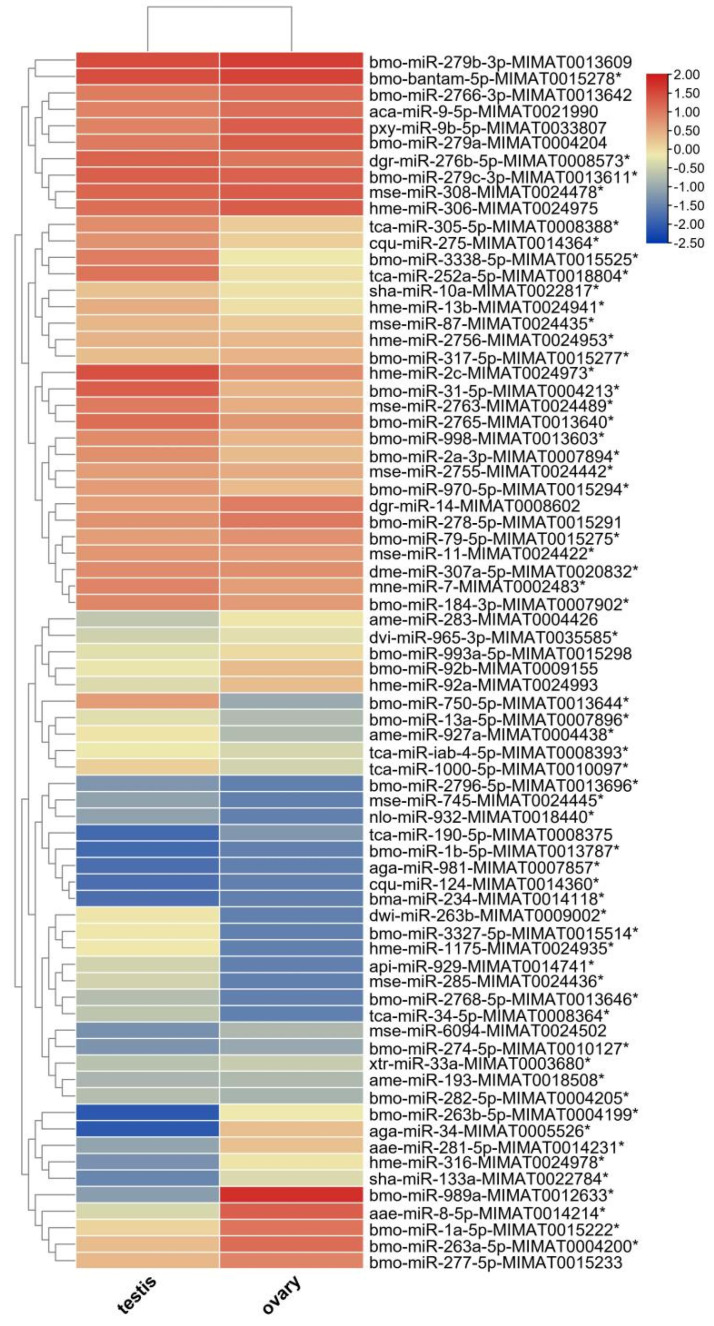
Expression levels of known miRNAs in the testes and ovaries of 0-day-old male adults and 2-day-old female adults of *H. armigera*. Putatively biased miRNAs (|log_2_FC| ≥ 1) between the testis and ovary are labeled with *.

**Figure 6 insects-12-00749-f006:**
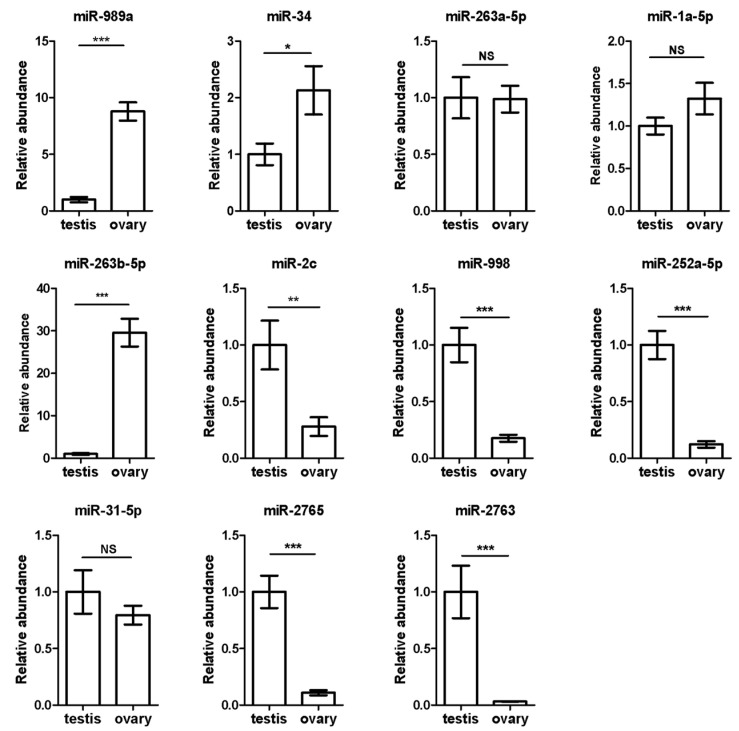
RT-qPCR validation of the expression levels of 11 ovary- and testis-biased candidate miRNAs in the testes and ovaries of 0-day-old male and 2-day-old female adults of *H. armigera*. U6 snRNA and let-7 were used as reference small RNAs to normalize the expression levels of the 11 gonad-biased candidate miRNAs. The data and error bars represent the means and standard errors of at least three biological replicates. The difference in the expression level of each miRNA between the testes and ovaries of sexually mature adults of *H. armigera* was analyzed by two sample *t*-tests. Significant differences at *p* < 0.05, *p* < 0.01, and *p* < 0.001 are depicted with *, ** and *** asterisks, respectively. NS = not significant.

**Figure 7 insects-12-00749-f007:**
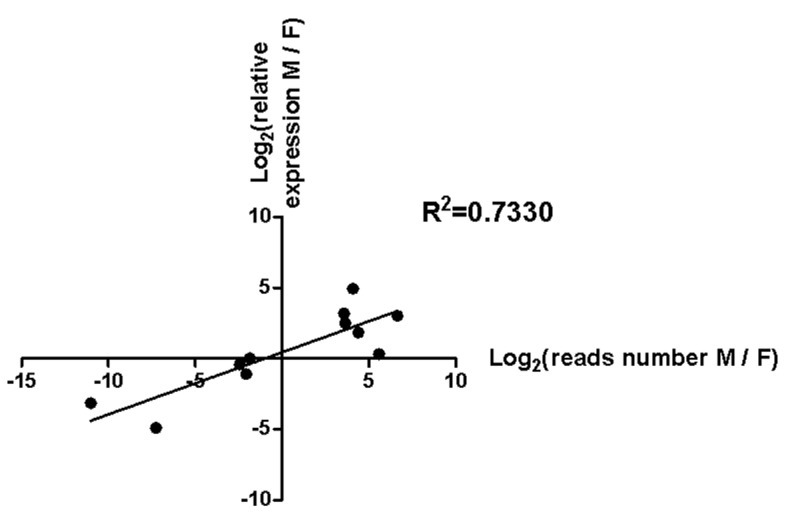
Correlation between the deep sequencing read data and RT-qPCR expression levels of the 11 testis- or ovary-biased candidate miRNAs.

**Figure 8 insects-12-00749-f008:**
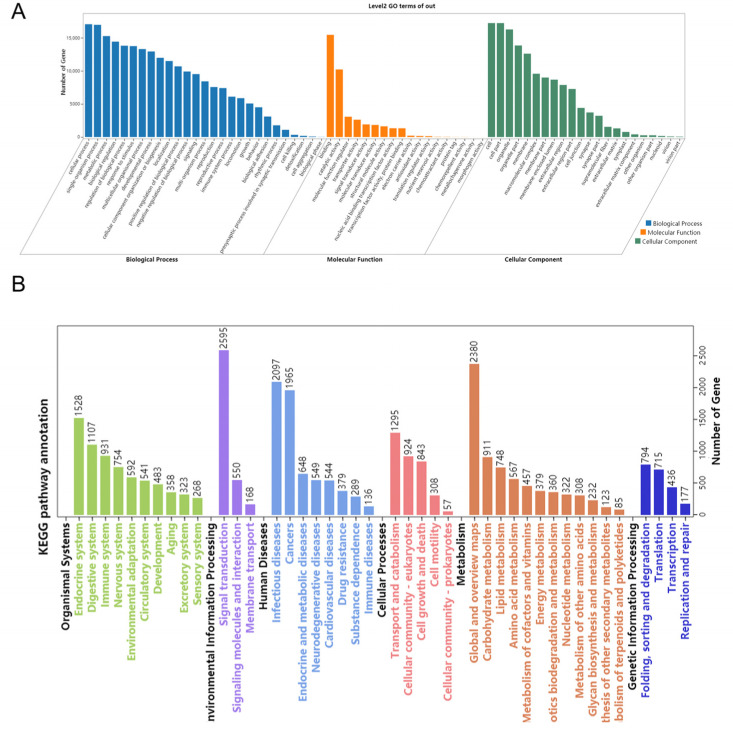
Gene ontology (GO) (**A**) and KEGG pathway (**B**) classifications of the 30,172 non-redundant putative target transcripts of the eight validated ovary- or testis-biased miRNAs.

**Table 1 insects-12-00749-t001:** Read counts and percentages of different types of small RNAs recovered from the ovary and testis small RNA libraries of *H. armigera*.

Ovary			Testis	
Type	Read Count	Percentage (%)	Read Count	Percentage (%)
Total	7,592,150	100	8,815,237	100
Unannotated	76,990	1.01	152,387	1.73
rRNA	842,672	11.10	1,379,525	15.65
snRNA	106,536	1.40	205,099	2.33
snoRNA	706,253	9.30	469,250	5.32
tRNA	312,634	4.18	649,231	7.36
miRNA	899,621	11.85	1,362,386	15.45
piRNA	2,611,968	34.40	1,012,824	11.49
Other	2,035,476	26.81	3,584,535	40.66

**Table 2 insects-12-00749-t002:** Numbers of putative target transcripts of the eight validated testis- or ovary-biased miRNAs.

miRNA	Number of Putative Target Transcripts
miR-998	4978
miR-989a	10,670
miR-34	12,041
miR-2c	3194
miR-2765	2239
miR-2763	324
miR-252a-5p	2134
miR-263b-5p	5474
Total	41,054
Shared by 2 or more miRNA	10,882
Total—shared	30,172

**Table 3 insects-12-00749-t003:** The reproduction-related pathways in which some of the putative target transcripts of the eight validated ovary- or testis-biased miRNAs participate.

KEGG_A_Class	KEGG_B_Class	Pathway	Transcript Number	Pathway ID
Cellular Processes	Cell growth and death	Oocyte meiosis	411	*ko04114*
Environmental Information Processing	Signal transduction	mTOR signaling pathway	653	*ko04150*
		JAK-STAT signaling pathway	166	*ko04630*
Metabolism	Lipid metabolism	Steroid hormone biosynthesis	285	*ko00140*
	Metabolism of terpenoids and polyketides	Insect hormone biosynthesis	62	*ko00981*
Organismal Systems	Endocrine system	Insulin signaling pathway	711	*ko04910*
		GnRH signaling pathway	385	*ko04912*
		Ovarian steroidogenesis	100	*ko04913*

## Data Availability

Small RNA sequences have been submitted to the SRA (Sequence Read Archive) database of NCBI with accession number PRJNA613606.

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
