# Peer review of "Identification and Characterization of MicroRNAs in Gonads of Helicoverpa armigera (Lepidoptera: Noctuidae)"

_insects, 2021, doi:10.3390/insects12080749_

Round 1

Reviewer 1 Report

Micro-RNAs have been shown to play an important role in regulation of insect reproduction and can be used as target in the development of novel insect pest strategies. Authors here report on the characterization and identification of miRNAs in the gonads of a serious lepidopteran insect pest, the cotton bollworm. 

qRT-PCR resulted in the detection of 60 known miRNAs in the ovary and 72 in the testis. They further showed that 8 gonad-biased miRNAs play an important role in reproduction and may be used as target for pest control in H. armigera and other lepidopteran pests.

The experiments are carefully done and the results are well presented. I have only minor points for improvement of the manuscript:

  • line 153 and others: give common names for animals in lowercase letters (human, rat etc.)
  • line 226: sentence is not complete (miRNAs?)
  • Legend to Fig. 1: give bar length in I to K
  • Fig. 3: delete the numbers above the columns
  • line 293: 20E is also a steroid hormone. Rephrase this sentenmce
  • Ref. 30: has to be corrected
  • Ref. 41: use the correct acronym for the journal title.

Reviewer 2 Report

The manuscript provides the results of a study characterizing MicroRNAs in gonads of a very important lepidopteran pest, Helicoverpa armigera.

This study adds information on previous knowledge based on model species such as Drosophila. Methods used to obtain the results presented are sufficiently described.

Based on the statement of authors on the utility of such characterization for the management of this pest, they should provide more concrete explanation of how possible pest management tactics could be develeped form their results.

Author Response

This manuscript is a resubmission of an earlier submission. The following is a list of the peer review reports and author responses from that submission.

Round 1

Reviewer 1 Report

MicroRNAs (miRNA) regulate multiple physiological processes including development and metamorphosis in insects. Here, authors describe the presence and functional role of miRNAs in the female and male gonads of a noctuid lepidopteran species.

The experiments were thoroughly done and the paper is well written. The manuscripts needs only minor corrections:

  • line 3, title: the correct name of the species is Helicoverpa armigera
  • line 19: miRNAs participate
  • line 28 and others: either use "Lepidoptera" or "lepidopteran"
  • line 71: differentially expressed
  • line 106: the strain was obtained
  • line 115: total RNA was extracted
  • line 155 and others: use kJ instead of kcal as IU for energy values
  • line 226: our data are reliable
  • line 248: authors have to say which "insect hormone" they mean (JH, E, 20E, neuropeptide?)
  • line 257, legend to Fig. 2: authors should say again at which day RNA lengths were measured (legend to figures and tables should be self explanatory)
  • line 268, legend to Fig. 4: the correct color is brown or orange, but not yellow
  • line 301: evidence suggests
  • line 320: again, you have to say which insect hormone is meant here
  • line 326: Corpora allata is the correct name
  • line 328: use uppercase letters for the insect orders (Diptera etc.)

Reviewer 2 Report

This study provides useful information on RNAs expression in H. armigera gonads, which improves our knowledge in the field.

Some work to improve language is necessary. Some sentences need to be rephrased.

Some examples (not an exaustive list) of the changes required is below:

L57: change "have" to "has" 

L88-94: too long sentence

L115: change "is" to "was"

L162: "...and the data were com-162 pared a two sample t-test with thresholds...." REPHRASE

Output of statistical analysis should be provided in all figures (histograms)

Round 2

Reviewer 2 Report

Authors have improved the manuscript according to reviewers' observations.

Still some spelling mistakes.

L.24: gene ontology (GO)
